# Machine-learning approach expands the repertoire of anti-CRISPR protein families

Ayal B. Gussow [1], Allyson E. Park[2], Adair L. Borges[2], Sergey A. Shmakov[1], Kira S. Makarova [1], Yuri I. Wolf [1], Joseph Bondy-Denomy[2] & Eugene V. Koonin [1]✉

The CRISPR-Cas are adaptive bacterial and archaeal immunity systems that have been harnessed for the development of powerful genome editing and engineering tools. In the incessant host-parasite arms race, viruses evolved multiple anti-defense mechanisms including diverse anti-CRISPR proteins (Acrs) that specifically inhibit CRISPR-Cas and therefore have enormous potential for application as modulators of genome editing tools. Most Acrs are small and highly variable proteins which makes their bioinformatic prediction a formidable task. We present a machine-learning approach for comprehensive Acr prediction. The model shows high predictive power when tested against an unseen test set and was employed to predict 2,500 candidate Acr families. Experimental validation of top candidates revealed two unknown Acrs (AcrIC9, IC10) and three other top candidates were coincidentally identified and found to possess anti-CRISPR activity. These results substantially expand the repertoire of predicted Acrs and provide a resource for experimental Acr discovery.

[1] National Center for Biotechnology Information, National Library of Medicine, National Institutes of Health, Bethesda, MD 20894, USA. [2] Department of Microbiology and Immunology, University of California San Francisco, San Francisco, CA, USA. ✉email: koonin@ncbi.nlm.nih.gov

All life forms evolve under constant pressure from numerous viruses and other parasitic genetic elements, and thus have evolved multiple defense systems[1]. The CRISPR-Cas are adaptive immunity systems that are present in nearly all archaea and ~40% of bacteria, and have been harnessed for the development of powerful genome editing and engineering tools[2–4]. In the incessant host–parasite arms race, viruses evolved multiple anti-defense mechanisms including diverse anti-CRISPR proteins (Acrs) that are currently known to comprise 46 distinct families[5,6]. The Acrs employ different mechanisms to abrogate the activity of CRISPR-Cas systems[7–10]. Most of the Acrs that have been studied to date bind to functionally important sites of CRISPR-Cas effector proteins and display high specificity toward a particular CRISPR-Cas variant from a narrow range of bacteria or archaea. Some Acrs, however, have broader specificity[11], for example, acting as nucleic acid mimics[12]. Furthermore, recently, enzymatically active Acrs, such as acetyltransferases and nucleases, have been discovered[13–15]. Clearly, Acrs have enormous potential for application as modulators of genome editing tools[16,17]. Despite the major interest of Acrs for understanding the biology of the host–parasite interactions in prokaryotes and their potential to transform the use of CRISPR in DNA editing, the discovery of Acrs remains a formidable task. The amino acid sequences of Acrs are extremely variable, which conceivably reflects the high variability and diversity of the CRISPR-Cas systems in bacteria and archaea[2]. The combination of the small size and the high evolutionary variability of the Acrs hampers their detection with even the most powerful sequence analysis methods[10]. The currently known Acr families were discovered using a variety of customized approaches, the two primary bioinformatic ones being guilt-by-association and self-targeting[5,12,18–20].

Guilt-by-association involves searching for homologs of HTH-containing proteins that are typically encoded downstream of Acrs[18]. Such proteins are known as anti-CRISPR associated (Aca) and are notably more conserved among viruses than Acrs themselves, which greatly facilitates their detection. The genomic neighborhoods encoding Aca homologs are then searched for potential Acrs.

Prokaryotic genomes containing CRISPR-Cas systems that encompass spacers targeting regions of the same genome are known as self-targeting[20]. In this case, CRISPR-Cas system should, in theory, target and kill the host cell. Therefore, organisms with self-targeting genomes can only survive when they also carry Acrs to prevent CRISPR-Cas from functioning (or, perhaps, by employing an alternative strategy for keeping the CRISPR-Cas silent) and thus keep the cell viable.

Despite the notable success of these two approaches, buttressed by experimental validation of many predictions, neither provides a comprehensive methodology for detecting Acrs. In addition to their extreme sequence variability, Acrs share few distinguishing characteristics outside of their common role in thwarting CRISPR. Here, we describe a systematic machine-learning approach we developed to predict Acrs, based on the few known Acr attributes and a secondary screen using heuristics of known Acrs, to further enrich for Acr candidates. We show that this method is significantly predictive of Acrs, compile a collection of 2500 previously undetected predicted Acrs families and examine the top candidates in detail, including experimental validation.

## Results

**Characteristic features of the known Acrs.** The general concept behind our approach is to combine the few characteristics Acrs tend to share into a detection model. Our first step was therefore to assemble and quantify features that previously discovered Acrs appear to have in common. To keep track of the known Acrs, we relied on a combination of curated Acr databases[21,22], and our own manual data curation (Supplementary Table 1). At the time of our data curation, 39 Acr families were known (Supplementary Table 1). We used this original set to iteratively search for homologs in the nonredundant (NR) database at the NCBI using PSI-BLAST and to construct a multiple protein sequence alignment for each Acr family.

We then used each of these alignments as the query for a PSI-BLAST search against our local protein sequence dataset[23] that includes prokaryotic and prokaryotic virus proteins, and consists of a total of 182,561,570 proteins. All hits with an e-value below the threshold of $10e-4$ were manually curated to eliminate obvious false positives, such as partial false-positive hits to very large proteins or hits to proteins with unambiguously assigned functions (unrelated to anti-CRISPR activity), in an effort to create a high-confidence Acr set. The final positive set consisted of 3654 Acrs, spanning 32 families (seven of the known Acr families were not represented in our database; Supplementary Table 1, Supplementary Data 1).

The most striking and obvious common feature of the Acrs is their small size (weighted mean Acr length: 104 aa, Table 1), and the tendency to form sets of small proteins that are encoded by co-directional and closely spaced genes in (pro)virus genomes (hereafter directons; Fig. 1, Table 1). We hypothesize that these directons are largely made up of co-transcribed early anti-defense genes.

Beyond these distinctive features, we considered other protein characteristics that we suspected might be predictive, such as the spacing of protein-coding genes within a directon ("Directon Spacing", Table 1) or protein hydrophobicity[24]

**Table 1 Feature set. Weighted means of all assessed features, and whether they were used in the final model.**

| Feature name | Acr mean | Non-Acr mean | Used in final model |
|---|---|---|---|
| Containing Genome is Prokaryote | 0.8956 | 0.8958 | No |
| Containing Genome is Self-Targeting | 0.3344 | 0.1919 | Yes |
| Directon Annotated Protein Fraction | 0.22 | 0.69 | Yes |
| Directon Protein Lengths Mean | 119.27 | 251.71 | Yes |
| Directon Predicted Membrane-Associated Fraction | 0.06 | 0.26 | No |
| Directon Size | 3.49 | 3.5 | Yes |
| Protein is Annotated | 0.0641 | 0.6731 | Yes |
| Protein has HTH-Downstream | 0.4008 | 0.1181 | Yes |
| Protein is Predicted Membrane Associated (TMHMM, SignalP) | 0.0256 | 0.2781 | No |
| Directon Spacing | 18.37 | 13.7 | No |
| Protein Length | 104.11 | 245.54 | Yes |
| Mean Hydrophobicity (Kyte and Doolittle) | −0.48 | −0.15 | Yes |

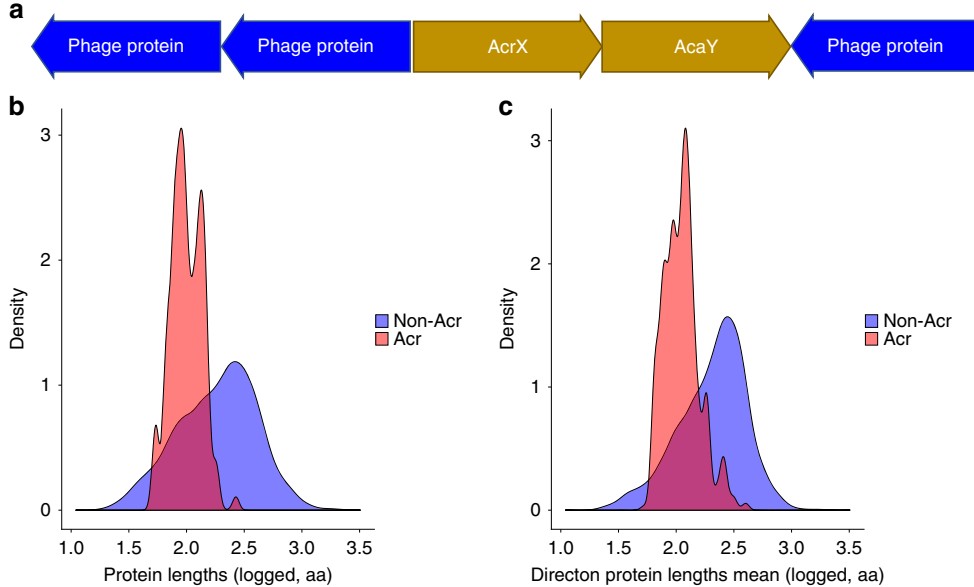

**Fig. 1 Characteristics of known Acrs. a** A cartoon of a sample directon. Acr proteins characteristically fall upstream of an HTH-domain-containing gene, termed Aca. Acrs are usually found in suspected mobile genetic elements, such as phages. The Acr directon is highlighted in the gold color, while the surrounding proteins are indicated in blue. Characteristically, Acrs fall in directons with small, unidentified proteins. **b** A density plot of Acr lengths. The $X$-axis denotes the common logarithm of the protein length, in amino acids. The $Y$-axis denotes the probability density function estimated from the data across the values of $X$. **c** A density plot of the mean lengths of proteins in Acr directons. The $X$-axis denotes the common logarithm of the mean length of proteins in an Acr directon, in amino acids. The $Y$-axis denotes the probability density function estimated from the data across the values of $X$.

("Mean Hydrophobicity", Table 1). We also considered whether proteins had significant hits when searched against conserved domains from either the NCBI Conserved Domain Database (CDD)[25] or Prokaryotic Virus Orthologous Groups (pVOG)[26] using PSI-BLAST ($e$-value < $10e-4$, "Protein is Annotated", Table 1), with the expectation that proteins with conserved domains likely perform other functions and therefore are unlikely to be Acrs. In total, we constructed a set of 12 features (Table 1, see "Methods" section for details) that, together, provided a compendium of quantifiable features that were used to identify Acr candidates.

**Training and test sets**. To build a predictive model, a training set comprised of two components was required: a positive set, consisting of previously discovered Acrs, and a negative set, consisting of proteins confidently inferred not to be Acrs (non-Acrs). For the positive set, the Acrs were weighted by their family and interfamily similarities (Supplementary Data 1, see "Methods" section for details), to ensure that related and highly similar Acrs were not overrepresented in the training dataset.

Because there is no well-defined, standard set of known non-Acr proteins, we constructed the negative set by randomly selecting viral and prokaryotic proteins, under the assumption that the majority of proteins are non-Acrs. The negative training dataset was constructed by randomly selecting proteins from a combination of 1000 randomly selected prokaryotic virus genomes and 4000 randomly selected CRISPR-Cas-containing prokaryote genomes. Similar to the positive set, we sought to avoid oversampling particular protein families. Therefore, these proteins were clustered by sequence similarity, and for each cluster, a single representative was selected. We randomly selected 3500 proteins from this set to constitute the negative, non-Acr set.

During our work on the predictive model, an additional set of Acrs was published[27,28]. We incorporated these into our analysis as an unseen test set, i.e., a set of Acrs unavailable during the training stage that we could use to test our model against. Thus, our training set consisted of all known Acrs published before September 2018 (Supplementary Data 1; positive set: $n = 2775$, 26 families; negative set: $n = 2600$), and the test set consisted of the Acrs published after that date (Supplementary Data 1; positive set: $n = 879$ proteins, 6 families; negative set: $n = 600$ proteins).

**Building and evaluating a predictive model**. Given our relatively small positive set, we sought to identify a model that would tend toward low variance. Thus, we chose a random forest of extremely randomized trees[29]. As an ensemble method with a highly random component, it is less likely than other machine-learning approaches to overfit the training data, while allowing a nonlinear mapping of features to label data and complex feature interactions.

The model consisted of a random forest with 1000 decision trees. When training the model, each decision tree is built based on a random sampling of the training data. Each split in the decision tree is determined by randomly selecting multiple values across a random subset of the features, and then setting the values that minimize the likelihood of misclassification as the thresholds for the decision tree split. Thus, the final forest consists of 1000 decision trees, where each decision tree's leaf nodes correspond to members of the training set.

When using the model to assess a candidate protein, the candidate traverses each decision tree. Within each tree, it ends up in a leaf node that contains some mixture of Acrs and non-Acrs from the training set. The tree assigns the candidate a score that is equal to the fraction of Acrs in its leaf. The score assigned by the model is the mean of the scores across all 1000 trees.

Using the model and the training set we developed, we assessed the performance of the model by five iterations of threefold cross-validation. In each iteration, the model was trained on two-thirds of the Acr families, and capacity to predict the families that were left out was assessed. For each protein in the test set, we predicted the likelihood of a protein being an Acr using our random forest

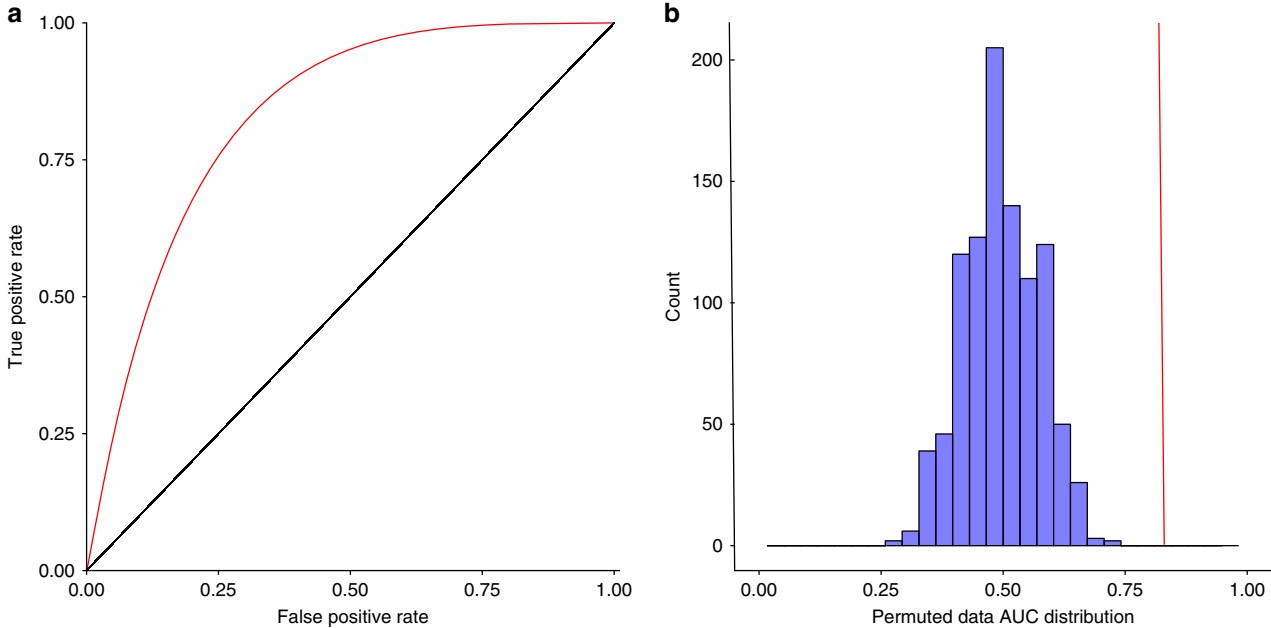

**Fig. 2 Model assessment on an unseen test set. a** The ROC AUC of the model scores on an unseen test set. **b** A histogram of 1000 AUCs calculated using permuted model scores against the unseen test set representing the null AUC distribution. As expected for a well-calibrated assessment, the null AUC distribution is centered on 0.5, indicating random separation. The AUC for the correct model scores, 0.83, is indicated with a red line.

model. Given the imbalance in the weights of samples in the positive and negative sets, we down weighted the negative set in training the model, so that its combined weight was equal to that of the positive set. This weighting was applied to both model training and assessment.

We relied on receiver operating characteristic (ROC) area under the curve (AUC) to assess the model performance and used a genetic algorithm for feature selection. The ROC is plotted based on the true-positive rates (the proportion of Acrs that are correctly identified) and the false-positive rates (the proportion of non-Acrs that are predicted as Acrs). On average, across all 15 cross-validation iterations, we found that our method was significantly predictive of Acrs, with an AUC of 0.93 (permutation $p$-value: 0.001).

We next used the model to predict Acrs in the unseen test set. The model was found to significantly distinguish Acrs from non-Acrs, with an AUC of 0.83 (permutation $p$-value: 0.001; Fig. 2). This result indicates that our method is indeed predictive of Acrs that are not present in the training set.

We converted the scores output by the model into binary predictions by setting a threshold for classification that maximizes the cross-validation balanced accuracy in the training set (Supplementary Fig. 1a). The binary model achieves a precision value of 78% and a recall value of 57% on the test set (permutation $p$-value: 0.001, Supplementary Fig. 1b, c). The members of three of the six Acr families assessed in the test set were detected most of the time (AcrIF12-IF14), whereas members of the remaining three families were detected less than half of the time, with the single member of AcrIE7 in the test set not detected by the model (Supplementary Fig. 1d, Supplementary Table 2).

**Using the model to predict Acrs**. Having formally demonstrated the predictive power of our model on the test set of recently discovered Acrs, we sought to leverage the model to generate a dataset that would be enriched for true Acrs. We combined the model predictions with other enrichment approaches based on known Acrs, under the expectation that this combination would

ultimately enrich for true Acrs, with the caveat that explicitly applying additional enrichment approaches skews the prediction performance away from that reported by the model, and might bias the resulting set to overlook Acr families that are distant from known Acrs.

First, we sought to define an appropriate search space of proteins likely enriched for Acrs. The initial dataset consisted of 182,561,570 proteins of that most (182,332,040) came from prokaryotes, and the rest were encoded by viruses (229,530). Acrs are typically encoded either within prokaryotic virus genomes, or within prokaryotic genomic regions that appear to be integrated viruses (proviruses) or other mobile genetic elements (MGEs)[10,19]. We therefore identified a subset of the prokaryotic database that consisted of genomes containing complete CRISPR-Cas systems[30], under the premise that these genomes are more likely to encompass prophages with Acrs targeting the respective CRISPR-Cas variants[16,20]. We further sought to limit the prokaryote protein set to proteins encoded by (putative) proviruses. Although there are many methods for predicting complete proviruses and their boundaries, these fall short of comprehensive identification of provirus regions in prokaryotic genomes, primarily, because numerous proviruses are inactivated and partially deteriorated[31,32]. Indeed, many of the known Acrs are encoded in the vicinity of virus proteins[12], but not necessarily within clearly active proviruses encoding hallmark virus genes and bounded by well-defined provirus boundaries. Therefore, instead of explicitly predicting proviruses, we enriched for virus-related sequence, by filtering the prokaryote protein set to the proteins encoded in the vicinity of known virus proteins (see "Methods" section for details). The resulting combined dataset of prokaryotic viruses and suspected proviruses consisted of 10,938,430 proteins. As these proteins are largely virus related, we expected this set to be enriched for Acrs.

This set of proteins was assessed with our random forest model that resulted in an initial set of 1,546,505 candidate Acrs. We further filtered these to retain only those that had no significant hits to CDD[25] or pVOG[26], yielding 892,830 proteins. This set of proteins was clustered by sequence similarity, resulting in 232,616

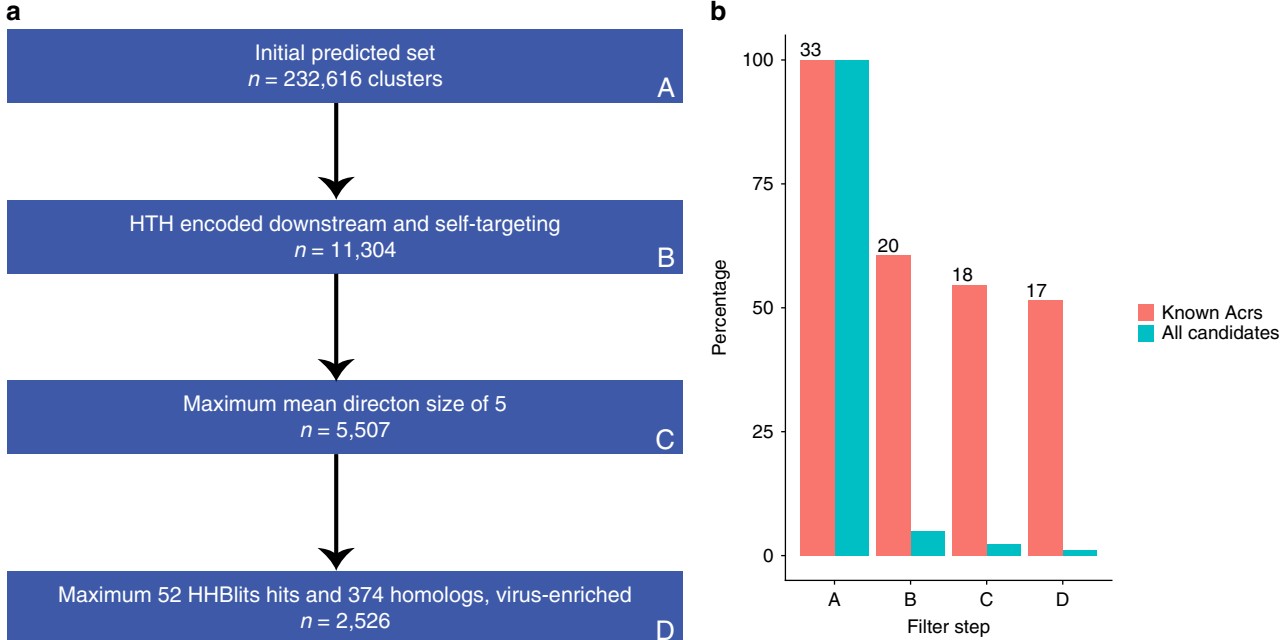

**Fig. 3 Heuristic filtering of the Acr candidates. a** Flowchart illustrating the heuristic filtering steps. The initial set consisted of 232,616 clusters and was first filtered for clusters that included at least one member with an HTH-domain-containing protein encoded downstream, and at least one member from a self-targeting genome, two hallmark Acr characteristics[20]. Four additional filters were applied, for mean directon size, number of HHBlits hits, number of homologs, and enrichment for virus homologs. The thresholds were set based on the data presented here. **b** Bar plot indicating the percentage of candidates and known Acrs from the initial positive set at each filtering step. The red bars denote the percentage of Acrs remaining at each step, and the blue bars denote the percentage of all candidates remaining at each step. The raw numbers of remaining known Acrs from the initial positive set are displayed above each red bar.

protein clusters (Supplementary Data 2). Heuristic filters were applied to each of these clusters, based on known Acr characteristics, to further enrich the candidate set for true Acrs. The hallmark characteristics of Acrs are that they (i) are encoded upstream of HTH proteins, and (ii) are found in self-targeting genomes[16]. We therefore required each family to have at least one member that fulfills each of these criteria. After this filtering, 11,304 families remained. Of these families, 20 included known Acrs from the initial positive set (Supplementary Table 3).

Following this filtering using the hallmark Acr characteristics, we developed and applied additional heuristic thresholds based on our initial observations. As genes encoding Acrs tend to form small directons, we sought to estimate a heuristic maximum threshold for the mean directon size in a candidate family that would enrich our protein set for true Acrs. We therefore searched for the threshold that, when applied, retained the largest fraction of the known Acrs in our set of 11,304, while filtering out as many of the candidate families as possible. To quantify this feature, we used the balanced accuracy metric, which is equal to the average of the fraction of correct classifications between the two groups. We found that a maximum mean directon size of five genes gave the highest balanced accuracy (see "Methods" section for details). Consequently, we removed protein families with an average directon size of more than five genes. After this filtering, 5507 families remained. Of the remaining families, 18 included known Acrs from the initial positive set (Supplementary Table 3).

To eliminate additional false positives, we performed a PSI-BLAST search of each protein family alignment against our sequence dataset and, under the premise that Acrs are highly variable, fast evolving proteins that are not known to be encoded outside the virus or provirus contexts, removed families with numerous homologs in diverse prokaryotes. We found that the heuristic cutoff value for the number of prokaryote homologs that

maximized the balanced accuracy was 374. We therefore limited our set to clusters with no >374 significant hits to the prokaryotic protein set. Next, we enriched for virus proteins by limiting to families that either include at least one homolog encoded in a virus genome or have a small ratio of prokaryote homologs to provirus homologs. We found that the cutoff value for the prokaryote to provirus ratio that maximized balanced accuracy was 3. Finally, we sought to exclude families that have numerous annotations when assessed with HHBlits and thus include well-characterized non-Acrs[33]. We found the cutoff value that maximized balanced accuracy for the number of HHBlits hits was 52. After this filtering, 2526 families remained. Of the remaining families, 17 included known Acrs from the initial positive set (Supplementary Table 3).

Although, by applying these heuristics, we likely filter out some true Acrs predicted by the model and bias our predictions toward the characteristics of known Acrs, we expect that, overall, this approach enriches the resulting protein set for true Acrs. After applying the above filters, our enriched set consisted of 2526 protein families (Fig. 3, Supplementary Data 2).

**Characteristics of predicted Acrs**. We performed a PSI-BLAST search of all 2526 candidate protein family alignments against a dataset of known Acrs and Acr-related sequences. For 26 of these families, significant hits to the Acr set were detected. Of these protein families, 22 included known Acrs. The remaining four families with significant similarity to known Acr-related sequences are homologous to uncharacterized proteins that are encoded within previously described Acr directons, namely, in the genomic neighborhoods of AcrIIA1-4 in *Listeria monocytogenes*, and all have been suspected of Acr activity although did not show such activity when tested[20]. These proteins were previously designated as OrfA, OrfB, and OrfE.

After removing these 26 families, we obtained 2500 candidate Acr families, consisting of 16,919 putative Acrs. The mean size of a family was seven, the largest family included 319 members, and nearly half of the families (49%) were singletons (Fig. 4).

Given the different cluster sizes, each predicted Acr was assigned a weight inversely proportional to the size of the respective cluster, in order to ensure that related and highly similar predicted Acrs were not overrepresented in summary statistics. Specifically, each predicted Acr was assigned a weight of $1/n$, where $n$ is the number of predicted Acrs in its cluster.

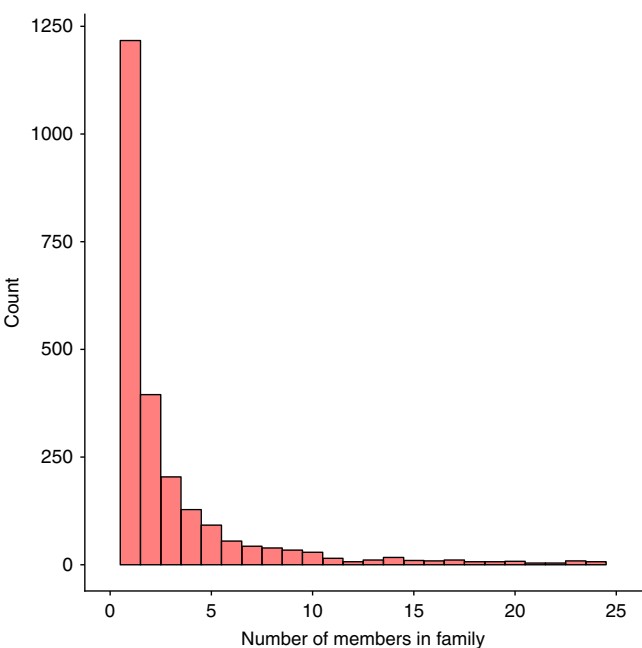

**Fig. 4 Histogram of predicted Acr family sizes.** For visual clarity, 134 families with >25 members were excluded. The families that are not displayed have a median of 52 members and a maximum of 319 members.

The predicted Acrs have a weighted average size of 109 aa, with a standard deviation (SD) of 71.6 (Fig. 5a). As expected by design, the Acr genes tend to form small directons (weighted mean: 3.4; weighted SD: 1.47) consisting of short genes (weighted mean of the protein sizes in the predicted Acrs directons: 200 aa; weighted SD: 155; Fig. 5b). The weighted mean isoelectric point of the predicted Acrs is 7.73 with a weighted SD of 2.6, and the weighted mean hydrophobicity is −0.31 with a weighted SD of 0.5. Per TMHMM and SignalP predictions[34,35], a weighted 15% of predicted Acrs have at least one putative transmembrane helix or signal peptide that, as expected, is substantially less than the expectation based on the negative set (28%, Table 1).

Using JPred[36], we predicted the secondary structure of the consensus sequences in the predicted Acr set. The mean percentage of amino acids contributing to alpha-helices was 39%, and the mean percentage of amino acids contributing to beta-sheets was 13%. In the negative set, 96% and 88% of the proteins were predicted to contain at least one alpha helix or beta sheet, respectively, and the mean percentage of amino acids contributing to alpha-helices and beta-sheets was 39% and 15%, respectively. Although these values do not differ substantially, we tested whether the distributions of the two categories differed significantly. We found a significant difference between the distributions of amino acids contributing to beta-sheets among the candidates and in the negative set (Mann–Whitney $U$ test $p$-value: 7.45$e$−13, Supplementary Fig. 2a), but no such difference for alpha-helices (Mann–Whitney $U$ test $p$-value: 0.823, Supplementary Fig. 2b).

The candidates are distributed across a diverse set of species ($n = 1,770$). *Escherichia coli* accounts for the largest share of candidate Acrs at 2.37%. *Peptoclostridium difficile* (1.46%) and *Clostridium botulinum* (1.16%) round out the top three. When considering how often each CRISPR-Cas subtype occurs in a genome containing a predicted Acr(s), including the cases of multiple subtypes, the three most common subtypes are I-E, I-C, and I-B (27.9%, 23.8%, and 22.2% of the genomes, respectively; Supplementary Table 4).

Among the 2500 candidate Acr clusters, 10% include at least one member encoded in a virus genome, with 279 virus strains

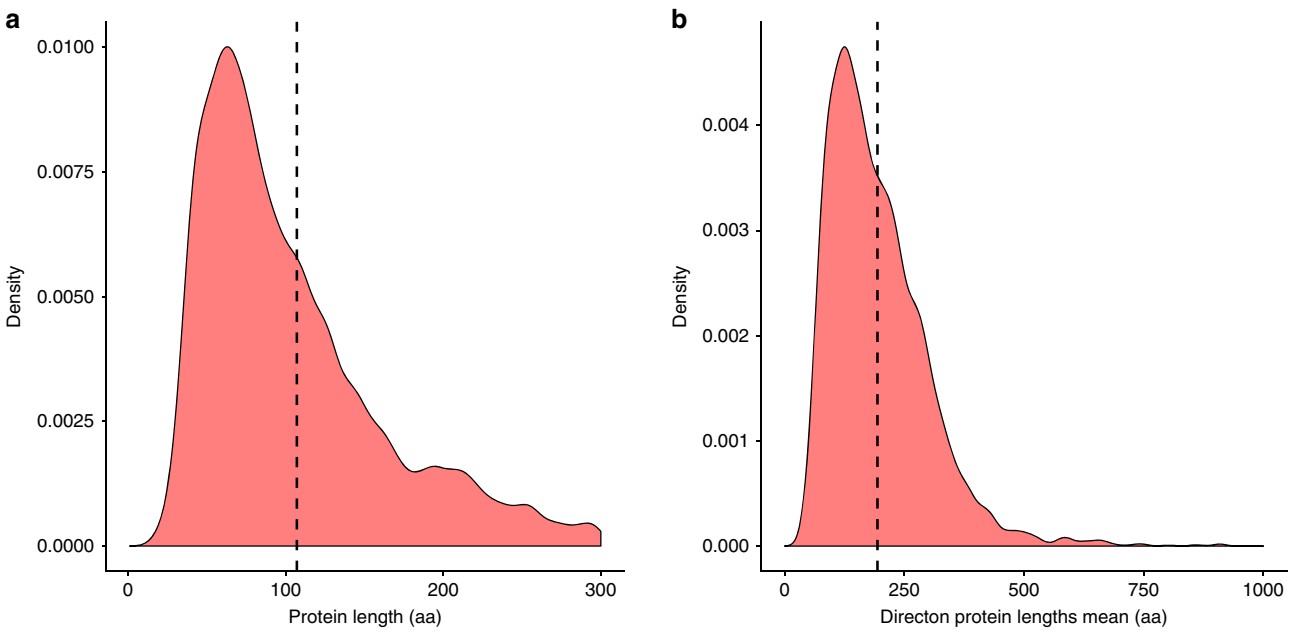

**Fig. 5 Protein length distribution of the Acr candidates. a** Density plot of the protein lengths of the predicted Acrs. The mean value (109 aa) is indicated with a dashed line. **b** Density plot of the mean protein lengths in the predicted Acrs directons. The mean value (200 aa) is indicated with a dashed line.

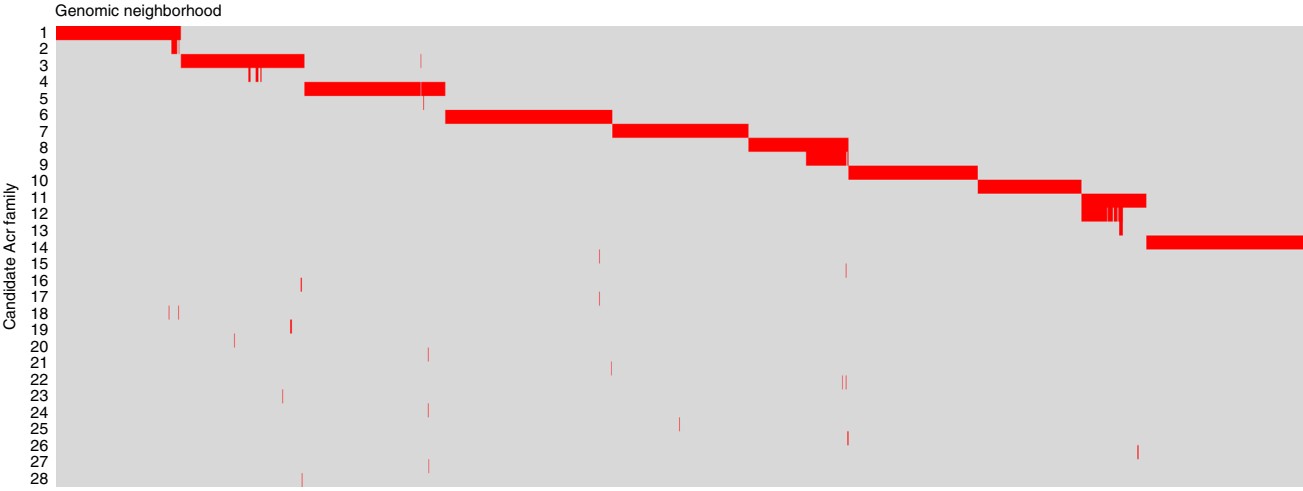

**Fig. 6 Presence–absence matrix of Acr families in genomic contexts.** A binary matrix where each column is a distinct genomic neighborhood (ordered by similarity) and each row represents an Acr family. Each cell represents the presence or absence of a member from the Acr family in the neighborhood, with red for presence and gray for absence.

encoding at least one Acr. Of the analyzed virus genomes, 197 (71%) encode a single predicted Acr, 66 (24%) encode two, and the remaining ones (5%) encode three or more Acrs. Archaeal viruses are also represented in this set, with 33 of the predicted Acrs found in 21 archaeal viruses.

The maximum number of predicted Acrs in a single virus strain was five, observed in Ruegeria phage DSS3-P1, four of which fell in the same HTH-containing directon. The viruses that were found to most commonly encode more than one Acr were *Mycobacterium* phages, followed by *Bacillus* and *Synechococcus* phages. Among the archaeal viruses, the viruses that were found to most commonly encode more than one Acr were *Sulfolobales* Mexican rudivirus followed by *Sulfolobus islandicus* viruses.

We sought to examine the genomic context of the largest predicted Acr clusters and gauge how often they tend to appear in similar genomic neighborhoods. We examined the ten largest Acr clusters and generated a presence–absence matrix for the members of these clusters in different genomic neighborhoods (Fig. 6), with a genomic neighborhood defined as the ten genes upstream and downstream of each Acr. Each column is a genomic neighborhood (ordered by similarity) and each row represents an Acr family. Whereas the larger Acr clusters in this subset tend to appear in similar genomic neighborhoods, within these neighborhoods, we also find scattered predicted Acr singletons. This pattern is similar to what has been observed in known Acrs, where the Acrs present in a given directon vary across closely related strains, with some Acrs appearing in nearly all instances of the directon and others appearing sporadically[20].

**Case by case analysis of top Acr candidates.** We next examined in greater detail the top candidates from our Acr candidate set. We constructed a set for in-depth examination, by filtering for clusters with more than four members and selecting the 30 clusters with the highest mean model score. These top 30 families were explored using HHPRED[37], PSI-BLAST against NR and examination of the genomic context for each candidate (Supplementary Data 3). Additionally, an overlapping but distinct set of 31 top candidates from Proteobacteria possessing associations with type I-C, I-E, or I-F were selected for experimental interrogation against these subtypes in *Pseudomonas aeruginosa* (see "Methods" section for details, Supplementary Table 5).

It has been previously shown that Acrs are typically encoded in short directons consisting of small genes, usually including one gene encoding an HTH-domain-containing protein[18,20]. This configuration has been observed for multiple Acr families and numerous virus and provirus genomes. One well-characterized example of this configuration involves the AcrIIA1-4 families[20]. Members of one of our top five candidate Acr clusters, candidate 4338 (hereafter C4338), were found in suspected prophages and phages of *L. monocytogenes*, adjacent to AcrIIA1, with three quarters of the members of this family found in self-targeting genomes. At the time of our analysis, C4338 was not found to be homologous to any of the previously discovered AcrIIA genes. However, shortly after the completion of the analysis and while this manuscript was in preparation, preliminary results on testing C4338 for an anti-CRISPR function have been reported independently[38]. C4338 has been identified as an anti-LmoCas9 protein (AcrIIA12), supporting the utility of our approach to discover Acrs.

Members of the C20391 cluster were identified in one phage (*Listeria* phage B054) and four suspected prophages (one in *Listeria innocua* and three in *L. monocytogenes*). All the prophage-encoded homologs were found in self-targeting genomes that carry CAS-II-A. Three of these genomes also carry CAS-I-B. All the prophage-encoded members of this cluster were found in bacterial genomes that also encoded AcrIIA1, and two of these also encoded Acrs IIA2 and IIA3. Given that all the genomes encoding proteins of this family encompass CAS-II-A, we predict that this is the target of its anti-CRISPR activity, although targeting of CAS-I-B is difficult to rule out.

As is characteristic of known Acrs, C20391 homologs are typically encoded in short directons consisting of three genes. One of these genes contains an HTH domain and is homologous to OrfD of *L. monocytogenes*. OrfD has been previously identified as a marker for Acr directons and is a distant homolog of AcrIIA1 although in itself, this protein has not been shown to possess Acr activity[20]. All members of this cluster are encoded adjacent to members of another predicted Acr family, C12805. C12805 includes three additional members that are not adjacent to C20391, but are all found in a directon with AcrIIA4 and an additional candidate, C42626, in prophages of *Listeria* strains solely containing CAS-I-B.

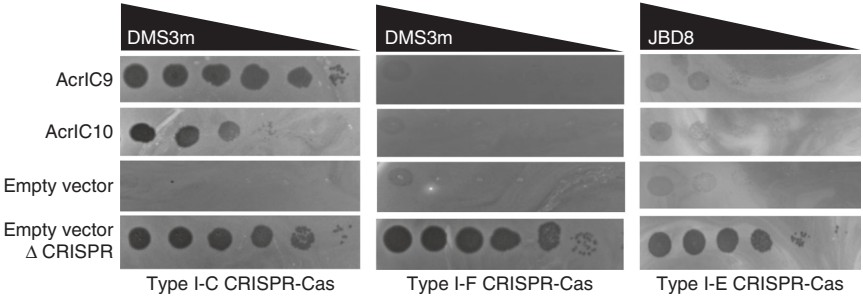

**Fig. 7 Identification of anti-CRISPR proteins AcrIC9 and AcrIC10.** Phage DMS3m or JBD8 spot titration from left to right (tenfold serial dilutions) on lawns of *P. aeruginosa* strains expressing the indicated CRISPR-Cas system (*X*-axis, type I-C, I-F, or I-E), with a crRNA targeting the indicated phage and the indicated Acr or empty vector (*Y*-axis). During screening of all candidate Acr proteins against all three systems, each combination was screened once. Upon detecting inhibition, positive results were visually confirmed in triplicate.

In the genomic neighborhoods of C42626, one instance includes an expanded version of AcrIIA4 (encoded in *L. monocytogenes* L99) that contains an HTH, whereas the remaining two instances of AcrIIA4 lack an HTH domain. However, an examination of the nucleotide sequences immediately upstream of the truncated AcrIIA4 indicates that this truncation is likely to be an error in the sequence annotation, and that the N-terminal of these instances of AcrIIA4 can be extended to match the AcrIIA4 homolog in *L. monocytogenes* 99, including the HTH domain. Furthermore, the region of AcrIIA4 that contains the HTH domain is similar to the portion of OrfD that contains an HTH domain (38% identity), so that extended version of AcrIIA4 appears to be a fusion of OrfD and AcrIIA4.

Thus, candidates C20391, C12805, and C42626 all contain the hallmark characteristics of known Acrs, including their tendency to fall in known Acr neighborhoods and next to known Acr markers. This corroborating evidence greatly raises our confidence that these are true Acrs and further validates the predictive power of the methodology.

Members of the C23907 cluster, experimentally validated as AcrIC9 (see below), were identified in one phage (*Rhodobacter* RcapNL) and in three RcapNL prophages integrated in self-targeting genomes of *Rhodobacter capsulatus*. C23907 belongs to a small directon of three genes, with the second gene in the directon containing an HTH domain. This HTH-containing gene is a distant homolog of Aca3, a previously discovered gene associated with Acrs, further supporting the prediction of anti-CRISPR functionality of C23907. The third gene in the directon is uncharacterized. The three self-targeting prophages containing the Acr occur in genomes with two CRISPR systems, type I-C and type VI-A, either of that are potential targets of C23907.

Members of the C27905 cluster were found in *Clostridium*. Half of the homologs were found in genomes that are self-targeting. As is characteristic of Acrs, C27905 genes typically belong to a small directon of 2–4 genes, where the second protein encoded in this directon contains an HTH domain. The other proteins in the directon are uncharacterized. All the genomes in this set contain CRISPR I-C, a potential target of C27905.

Members of the C11640 cluster, experimentally validated as AcrIC10 (see below), were found in *Xanthomonas*. Eight of these genes were identified in *Xanthomonas translucens* and one in *Xanthomonas* sp. SHU199. Eight of the nine homologs were found in self-targeting genomes. C11640 tends to fall in a small directon of two genes, as is characteristic of known Acrs, where the second gene in the directon contains an HTH domain. All the genomes containing C11640 have type I-C CRISPR systems, a potential target of C11640.

To test these predictions, Acr validation was conducted in *P. aeruginosa* strains expressing type I-C, I-E, or I-F CRISPR-Cas systems targeting phage (see "Methods" section for details). Type I-E and I-F systems were expressed at endogenous levels with native spacers targeting phages JBD8 and DMS3m, respectively, whereas the type I-C system was expressed heterologously in strain PAO1, with an engineered spacer. The candidates associated with one of these three subtypes and present in *Pseudomonas*, combined with members of the top 30 Acr set (Supplementary Data 3) from Proteobacteria, were selected for experimental validation. Two candidates identified in this work (C40699 and C25827, Supplementary Table 5) were found to be homologous to two type I-C CRISPR inhibitors that have been independently identified via Aca association[39]. C25827 is homologous to AcrIC3 (100% amino acid identity, 100% coverage), and C40699 is homologous to AcrIC4 (89.4% amino acid identity, 98% coverage). The remaining 29 genes were synthesized and 23 were successfully cloned into expression vectors (Supplementary Table 5). In addition to AcrIC3 and AcrIC4, anti-CRISPR activity was demonstrated for two proteins (AcrIC9 and AcrIC10) that also targeted the type I-C system (Fig. 7, Supplementary Table 5). AcrIC9 is 79 amino acids in length and highly acidic (pI = 3.96), much like previously described DNA mimic Acr proteins[9,40]. This protein was found to be highly active, fully inactivating type I-C CRISPR-Cas. AcrIC10 is 94 amino acids and is a comparatively neutral protein (pI = 6.57) that displayed an ~100-fold weaker activity than AcrIC9, under our experimental conditions.

AcrIC9 and IC10 were the two highest confidence candidates tested, and AcrIC4 and IC3 were the fourth and sixth highest confidence candidates tested, respectively. Of the remaining 21 candidate Acr proteins, four were toxic in the type I-C strain and therefore were not tested against I-C, and only six were tested against type I-F before the laboratory shutdown due to COVID-19 (Supplementary Table 5). Together, these results confirm that top-ranking predictions by our method are highly likely to be active Acrs.

## Discussion

The Acrs are of major interest to a wide range of researchers, due both to their role in the evolutionary arms race between viruses and their prokaryotic hosts, and to their potential use as CRISPR-Cas inhibitors in genome engineering applications. Here, we demonstrate substantial predictive and discriminative power of a machine-learning approach for the identification of candidate Acrs. This result appears unexpected given the paucity of distinctive features of the Acrs. Nevertheless, these few, rather generic features including the small size of the Acr genes,

their arrangement in short directons that contain, additionally, genes for HTH proteins, poor evolutionary conservation, association with viruses and proviruses, and self-targeting seem to be sufficient for apparently robust Acr prediction. The underlying reason seems to be that, in viruses of prokaryotes, a substantial fraction, often, the majority of the genes that are not directly implicated in virus replication and morphogenesis are involved in anti-defense functions. A notable example can be found among archaeal viruses in some of which up to 40% of the genes appear to encode Acrs[41]. Hence a possible caveat of our predictions: some of the genes that we predict as Acrs might target other, non-CRISPR defense systems. Conversely, the possibility exists that, using the approach described here, we only detect one, albeit major, class of Acrs, whereas others might exhibit distinct properties.

The above caveats notwithstanding, the combination of sensitive database searches, machine-learning and heuristic filters applied here yielded 2500 previously undetected families of strong Acr candidates that comprise an extensive resource, which we make accessible online (http://acrcatalog.pythonanywhere.com/), for structural and functional studies on Acr-CRISPR interactions, with likely subsequent applications. The experimental validation presented here and elsewhere confirmed many of the top predictions. Genes that tested negative for CRISPR-Cas inhibition against single representatives of type I-C, I-E, and I-F in *P. aeruginosa* could lack inhibitory activity in this assay for many reasons. They might be Acrs specific for different variants of the tested subtypes or different subtypes altogether, or Acrs that act at a different stage of immunity, such as spacer acquisition. The three model strains used to represent the type I-C, I-E, and I-F systems do not necessarily reflect the potential interactions between the candidate Acrs and diverse variants of these systems, or different CRISPR-Cas types and subtypes present in the genomes where the Acr candidate was found. Future work will be required to test these candidates against relevant systems in the species of interest. Lastly, some of these candidate Acr proteins might inhibit other, non-CRISPR-based bacterial immune systems, given that, as recently shown, anti-defense genes show a strong tendency to cluster in MGEs[42]. The signatures of Acrs described in this work might apply broadly to inhibitors of other prokaryotic immune systems.

The current database of prokaryotic virus genomes is limited in scope but grows rapidly, thanks, largely, to metagenomic discovery of numerous viruses[43]. Furthermore, so far, no targeted search for Acrs in MGEs other than viruses, such as plasmids or transposons, has been performed. Characterization of the distribution of Acrs throughout the prokaryotic mobilome is a key next step to understanding the arms race that can be expected to lead to the discovery of numerous Acrs. Thus, the clear extension of this work involves searching the expanding virus genome databases, metagenomes, and other MGE. Iterative application of this strategy should greatly expand the diversity of Acrs and, possibly, inhibitors of other defense systems.

## Methods

**Iterative search for Acr homologs.** For each Acr family, a single representative sequence was selected, and a PSI-BLAST search was run against the NCBI NR sequence database. Iterative PSI-BLAST was run to convergence, the identified homologs were aligned using MUSCLE[44], and the resulting alignment was searched against our prokaryote dataset[23] and our prokaryotic virus dataset from the NCBI viral genomes resource[45], using PSI-BLAST. We used a cutoff of e-value ≤ $10e-4$ for homolog detection and manually reviewed each resulting alignment. Of the 39 assessed families, seven were not detected in our database (Supplementary Data 1). As the database used in this study was curated in 2016 (ref. [23]) and does not include all known proteins, and because Acrs tend to be highly variable, with few homologs, it was not unexpected that these families were missed. All seven families not in our database were originally detected in strains that were not available at the time the database was constructed.

**Weighting the Acrs.** For the positive set, we sought to weight each Acr by its sequence similarity to the other Acrs, in order to avoid oversampling closely related data points. Initially, each Acr family is assigned a weight of one. Then, within each Acr family, its member proteins were clustered using mmseq2, with the parameters $c = 0.5$ and $s = 0.4$ (ref. [46]). Each cluster is defined as a subfamily, and the initial weight of one given to the family is divided evenly amongst the subfamilies. Following this, each subfamily's weight is divided evenly among its members. Thus, each Acr's weight is proportional to its similarity to other Acrs in the set.

For the negative set, an analogous procedure was followed. After randomly selecting a set of proteins as the negative set pool, these proteins were clustered using the same mmseq2 parameters as used for the Acr families, and from each cluster, a single representative was selected. Each representative protein was given a weight of one.

In training and in assessing the model, the negative set was reweighted so that each class (Acr and non-Acr) had the same total weight.

**Protein annotations.** Proteins in our dataset were annotated by applying a PSI-BLAST search against CDD[25] and pVOG[26], with an e-value cutoff of $10e-4$. Proteins with hits to pVOGs were classified as viral. When enriching for true Acrs using heuristics, proteins with hits to either CDD or pVOG were eliminated.

**Self-targeting assemblies.** Self-targeting assemblies were detected by BLASTing the spacers[30] from each assembly against the corresponding genome and filtering for exact matches. Wherever an exact match was found, the respective assembly was classified as self-targeting (Supplementary Data 4).

**Defining the features for the model.** Overall, 12 total features were defined. Some features related to the protein itself, while others relate to the protein's directon. A directon was defined as consecutive proteins on the same strand with a maximum of 100 bp between adjacent proteins.

The features were defined as follows:

*Protein size*: The length, in amino acids, of the candidate protein.

*Directon size*: The number of genes in the directon.

*Mean directon protein size*: The mean length, in amino acids, of all proteins in the directon.

*Protein hydrophobicity*: The protein's hydrophobicity[24].

*Protein annotation*: A binary score of whether the protein is annotated or not. We consider a protein as annotated if it has at least one significant hit to any alignment, outside of alignments annotated as hypothetical protein, putative predicted product, or provisional.

*Fraction of directon that is annotated*: The fraction of proteins in the directon that are annotated as defined above.

*HTH-downstream*: Whether there is an HTH-domain-containing protein encoded downstream of and adjacent to (within three genes) the Acr candidate within the same directon. This feature was analyzed by running a PSI-BLAST search of proteins against the subset of alignments from the PVOG and CDD datasets containing in their name, or description either the term HTH or helix-turn-helix, with an e-value cutoff of $5e-3$.

*Self-targeting*: Whether the protein is encoded in a self-targeting genome.

*Predicted membrane association*: Whether the gene is predicted to be transmembrane or contain a signal peptide using TMHMM and SignalP, respectively[34,35].

*Fraction of membrane-associated proteins in directon*: The fraction of the proteins encoded in the directon that are predicted to be transmembrane or contain a signal peptide as defined above.

*Directon spacing*: The mean spacing between genes in the directon.

*Whether genome is viral*: Whether the protein is encoded in a viral genome or in a prokaryotic genome.

A genetic algorithm that selects subsets of features and creates different feature combinations while optimizing for the best feature set[47] was applied to the 12 features for ten generations, yielding the following feature set:

(1) Containing Genome is Self-Targeting
(2) Directon Annotated Protein Fraction
(3) Directon Protein Lengths Mean
(4) Directon Size
(5) Protein is Annotated
(6) Protein has HTH-Downstream
(7) Protein Length
(8) Protein Hydrophobicity

**Building the model.** The model was constructed using scikit-learn (https://scikit-learn.org), specifically, the ExtraTreesClassifier with the the the $n\_estimator$ parameter set to 1000, meaning that the random forest consisted of 1000 trees. The rest of the parameters were left at default. A random forest was chosen to model the data given that it is an ensemble classifier that is less likely to overfit than other methods, while allowing a nonlinear mapping of features to labels data and complex feature interactions[29].

The model was trained on the training dataset described above, while down-weighting the negative set so that each class (Acr and non-Acr) has the same total weight. The thresholds for each split in the random forest trees were selected to minimize Gini impurity[48], which measures how often misclassification would occur when a randomly selected member of the node is randomly classified based on the distribution of labels in the node, calculated as follows:

$$I_G(n) = 1 - \Sigma_{i=p}^{2}(p_i)^2, \qquad (1)$$

where $I_G(n)$ is the Gini impurity for node $n$, and $p_i$ is the fraction of samples for class $i$ (either Acr or non-Acr) in node $n$. Thus, the Gini impurity reaches 0 when all samples in the node fall into a single category.

Predictive scores were calculated by using the ExtraTreesClassifier function *predict_proba*. When calculating binary predictions, the threshold was set to the best value for differentiation in the training set when maximizing accuracy, which was equal to 0.09.

**Defining the Acr search space.** The alignments of the pVOG proteins were compared to the dataset of genomes containing CRISPR-Cas[23,30]. Each directon containing a protein with a viral hit with an *e*-value <10*e*−4 was considered a provirus-related sequence, along with the adjacent directons on either side. Adjacent blocks of prophage-related directons (within 500 bp of each other) were considered as provirus candidates. If the provirus candidate contained at least two virus hits within 3 kb of each other, it was considered a predicted prophage.

The set of virus proteins was assembled from the NCBI viral genomes resource[45], and subset to prokaryotic viruses based on taxonomy data (https://www.ncbi.nlm.nih.gov/genomes/GenomesGroup.cgi?taxid=10239). This virus set totaled 229,530 proteins encoded in 2291 genomes.

**Permutation *p*-value calculation.** To calculate permutation *p*-values, the model's predictions for the test set were shuffled. We then tested how well the model performed on this shuffled dataset. This procedure was repeated 1000 times, creating a null distribution of AUCs. With this null distribution, a permutation *p*-value was calculated as follows. Let $n_p$ be the number of AUCs in the null distribution that are greater than or equal to the actual observed AUC. The permutation *p*-value, then, is equal to $\frac{1+n_p}{1001}$. Thus, when the actual AUC was greater than any AUC in the entire permuted set, the *p*-value was ~0.001.

**Clustering and weighting candidate Acrs.** Candidate Acrs were clustered using mmseq2, with the parameters $c = 0.5$ and $s = 0.4$ (ref. [46]). A weight of $1/n_c$ was assigned to each cluster, where $n_c$ is the number of Acr candidate clusters. The weight of each cluster was then divided evenly among all protein members of the cluster, so that the weight of each Acr was inversely proportional to the size of the cluster it belonged to. These weights were used when calculating summary statistics for the Acr candidate set, to avoid oversampling closely related data points.

**PSI-BLAST search against known Acr and Acr-related sequences.** We created a sequence database of known Acrs and Acr-related sequences (Supplementary Data 5). This database included all known Acrs, Acas, and proteins previously suspected of possessing Acr activity, but not showing any when tested. We included the group of previously suspected Acr proteins as these are proteins that bear Acr characteristics, and therefore may be detected by our method, but have already been tested for Acr activity.

A PSI-BLAST search of each candidate Acr cluster alignment as the query was performed against this dataset of known sequences, the clusters that produced hits with an *e*-value of <10*e*−3 were discarded as belonging to known Acr families or families that have already been already tested for the Acr function.

**Heuristic filtering.** To choose the thresholds for all the heuristics except for self-targeting and HTH-downstream, ten evenly spaced threshold values were tested, between the minimum Acr value and the maximum Acr value. Each of these ten thresholds were applied as cutoffs to the Acr families, and for each threshold the balanced accuracy was calculated. The balanced accuracy is equal to the mean of the percentage of known Acrs that passed the threshold and the percentage of all proteins that were filtered by the threshold, so that a higher balanced accuracy corresponds to better discrimination between the known Acrs and the rest of the candidates. The final threshold was selected so as to maximize the balanced accuracy. The selected threshold was then applied to the dataset.

Six heuristics were defined to further enrich the Acr candidate set.

*Number of members that have HTH-downstream*: We required that at least one member of the candidate family have an HTH-containing protein encoded downstream within the same directon.

*Number of members in self-targeting or virus genome*: We required that at least one member of the candidate family was either encoded in a self-targeting genome or encoded in a virus genome.

*Mean directon length*: The mean number of genes in the directon for all members of the family.

*Number of homologs in prokaryotic dataset*: A PSI-BLAST search of the multiple protein alignment of each family was performed against the prokaryotic sequence dataset[23], and filtered for hits with a maximum *e*-value of 10*e*−6, 50% identity and 50% query coverage.

*Ratio of prokaryotic homologs to predicted provirus homologs*: A PSI-BLAST search of the multiple protein alignment of each family was performed against the predicted provirus sequence dataset and the virus sequence dataset, and filtered for hits with a maximum *e*-value of 10*e*−6, 50% identity and 50% query coverage.

If a family produced at least one hit to a virus sequence, it was included. If not, it was required that the ratio between the number of hits to the prokaryotic sequence dataset to the number of hits to the predicted provirus dataset was less than or equal to three.

*Number of HHblits hits*: The alignment of each family was compared to PFAM[49] and PDB70 (ref. [50]) using HHblits[33]. Families with >52 hits were discarded.

**Construction of Acr presence–absence matrix.** To generate the presence–absence table, for the ten largest Acr clusters, ten genes upstream and downstream were extracted where available (a maximum of 20 genes total). If within this set, an additional predicted Acr was represented, the set was further extended to include the ten genes upstream and downstream of that additional predicted Acr. The resulting gene arrays were considered the Acr genomic neighborhood.

A binary matrix was constructed where each column is a genomic neighborhood, ordered by content similarity, and each row is a predicted Acr family. In addition to the Acrs from the top ten largest clusters, those encoded within ten genes upstream or downstream of Acrs from the largest clusters were included. Each cell represents the presence or absence of a member of the respective Acr family in the neighborhood.

**Manual assessment of candidates.** The multiple alignment for each of the top 30 candidates in Supplementary Data 3 was compared against the PDB, PFAM, and NCBI CD databases using HHPRED[37]. For each candidate, we calculated a consensus sequence, where the consensus letter for an alignment position was defined as the amino acid that has the highest BLOSUM62 score among the amino acids occupying the position. A PSI-BLAST search of the consensus sequence of each candidate family was performed against NR, and the genomic contexts of homologs were visually assessed using Geneious Prime.

**Plasmid preparation.** All candidate proteins were reverse translated and codon optimized for *P. aeruginosa* PAO1 using IDT Codon Optimization Tool. Gene fragments (TWIST Biosciences) were cloned into the SacI/PstI site in the pHERD30T vector using Gibson Assembly. The resultant plasmids were selected with 30 μg/mL gentamicin and propagated in *E. coli* strain DH5α.

**Transformation.** All plasmids were transformed via electroporation into *P. aeruginosa* strains LL77 (a PAO1 derivative with the type I-C *cas* genes integrated in the chromosome), SMC4386 (native type I-E), and PA14 (native type I-F) to test for inhibition of the type I-C, type I-E, and type I-F CRISPR-Cas immune systems, respectively. Transformation was performed using *P. aeruginosa* cultures grown overnight in LB medium at 37 °C with shaking. To make cells electrocompetent, 1 mL of overnight culture was pelleted, resuspended in 1 mL of 10% glycerol, and then pelleted and resuspended twice more, with the final resuspension done with only 100 μL of glycerol solution. A total of 1 μL (~300 ng) of plasmid was added to the electrocompetent cells, and the mixtures were allowed to sit on ice for 30 min. The cells/DNA mixture was transferred to cuvettes and electroporated using the BioRad Gene Pulser Xcell Electroporation Systems preset *P. aeruginosa* setting. Immediately after electroporation, 1 mL of LB was added to each cuvette. The cells were transferred from the cuvettes to 1.5 mL eppendorf tubes and recovered for 1 h at 37 °C with shaking. The recovered cells were pelleted, the top 700–800 μL of supernatant was removed, and then the cells were resuspended in the remaining supernatant. A total of 150 μL of cells were then spread with glass beads onto LB agar plates with 50 μg/mL gentamicin. The plated cells were allowed to grow overnight at 37 °C.

**Plaque assay for CRISPR-Cas activity.** Single colonies of the three testing *P. aeruginosa* strains with the candidate plasmids were grown overnight in 3.5 mL of LB medium with 50 μg/mL gentamicin. A total of 150 μL of each overnight culture were then mixed with 3.5 mL of molten top agar (supplemented with 1 mM IPTG for LL77) in small glass tubes. The resultant agar–bacteria mixture was then poured onto circular LB agar plates with 50 μg/mL gentamicin, 0.1% arabinose, and 10 mM MgSO₄. After being left to dry for 10 min, tenfold serial dilutions of CRISPR-targeted bacteriophage, ranging from 1 to 10⁻⁶ were pipetted onto the plates, and the plates were then incubated overnight at 30 °C. Phages JBD30, JBD8, and DMS3m were used to assay type I-C, type I-E, and type I-F CRISPR-Cas activity, respectively. Plaque assays were conducted on standard petri plates, 10 cm in diameter.

**Reporting summary**. Further information on research design is available in the Nature Research Reporting Summary linked to this article.

## Data availability

The authors declare that the data supporting the findings of this study are available within the paper and its Supplementary Information files. The NCBI NR sequence database is available at https://ftp.ncbi.nlm.nih.gov/blast/db/ under the file names "nr.*.tar.gz". Other data are available from the corresponding author upon reasonable requests. Source data are provided with this paper.

## Code availability

The model code and sample data are available on GitHub (https://github.com/gussow/acr).

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

## Acknowledgements

The authors thank Koonin group members for helpful discussions. This research was supported by the Intramural Research Program of the National Library of Medicine at the NIH.

## Author contributions

A.B.G., K.S.M., Y.I.W., and E.V.K. designed research; A.B.G. performed research; A.E.P. and A.L.B. performed experimental tests; A.B.G., S.A.S., K.S.M., Y.I.W., A.E.P., A.L.B. J.B.-D., and E.V.K. analyzed data; and A.B.G. and E.V.K. wrote the paper.

## Competing interests

J.B.-D. is a scientific advisory board member of SNIPR Biome and Excision Biotherapeutics, and a scientific advisory board member and co-founder of Acrigen Biosciences.
