## [Peer Review File · Nature Communications]

REVIEWER COMMENTS

Reviewer #1 (Remarks to the Author):

In this work, Gussow et al present a new, machine-learning-based approach to bioinformatic prediction of anti-CRISPRs. The result is an enormous expansion of Acrs and Acr families. Furthermore, as these are fascinating (and commercially exploitable) proteins, the work is impactful and an important contribution to a hot and fast-moving field. It is thorough, too – catching tricky concepts like the error in sequence annotation for AcrIIA4s. Great work!

I have no critical or major concerns. The paper is well suited for the journal, presents a major advance in the field, and does not overextend its findings. However, I did find the paper challenging to read. While the flow of ideas was logical and the language used was appropriate, there was a lot of repetition of specific phrases that felt like padding, which detracted from the manuscript. I also found the Figures to be challenging to navigate, often for reasons with very simple fixes – and almost all could have used more detailed legends (and moved below the figures for legibility). I will begin by listing a few Moderate and Minor concerns and append a list of line-specific issues.

I do apologize for the emphasis on flow/wording. In this lockdown mode with 3 young children at home, my review time is limited and issues like these disproportionately stand out.

Moderate:

(1) This point concerns me throughout the manuscript, and it probably has more to do with my only passing familiarity with bioinformatics. The dataset was built eliminating hits to large proteins, but then the supposed striking feature of ACRs is their small size - how influenced is this by the bias inserted at that earlier stage? Many ACRs were identified next to Acas, so how surprise is it really that your Acr database also includes ACRs typically encoded upstream of Acas?

Later, you identified a subset of the prokaryotic database with complete CRISPR-Cas systems – is this not stacking your dataset for self-targeting spacers? Shouldn't the algorithm handle these and sort away the non-CRISPRs anyway (since self-targeting is weighted), rather than biasing the search dataset?

Then, you filter the set to remove those with significant (? Unclear on criterion) hits to CDD or pVOG. Again, this is one of your machine learning criteria anyway, so why do you need to bias the dataset?

As stated, I feel this is likely just a failure on my end to grasp some simple element, but as this feeling pervaded my entire readthrough of the manuscript, it may be important to address.

(2) You removed families with numerous homologs in diverse prokaryotes. I worry this biases against acrs in bacteria over-represented in the sequence databases? I doubt there is a real issue here, as its not like your cutoff of 375 was anything close to your largest family (319).

(3) I... think this may be a stupid question, but how many of your training set would survive your heuristics from Fig 3? Might this be worth having as a panel for Fig 3 as well? I am almost certain this is a stupid question.

(4) Is the resolving power between a 3.49 and 3.5 mean directon size really useful? That seems... wild.

(5) The figures need improvements throughout the manuscript. In some cases they should be omitted (Fig 7), most – more carefully labelled. Here are some specific points on the figures

5.1 Fig 1 axes are confusing – protein length is in what units? Are Direction Protein lengths Mean the average length of proteins in the directon? Presumably, but this could be clearer. Is density the correct Y label? I assume it is count?

5.2 Fig 2: why the red dot? Why is this not a line, a threshold... why is it at 0.5 on the y axis?

5.3 I think the paper might benefit from emphasizing the new criteria and heuristics – in the tables 1 and 2, for instance. IN table 2 the first 2 criteria are the ones everyone uses while the next 4 are new. I also think Figure 3 and Table 2 should be merged together

5.4 Fig 4: Legend should at least give us cursory info about families with more than 25 members (the X families with more than 25 members). Better axis labels; Size of family (# of members)

5.5 Fig 5: show mean on figures. Show units (aa)

5.6 Fig 6 is hard to read. Grey in legend is white? I bet you can do more with this figure.

5.7 Fig 7 does not feel like it should be its own figure. Isnt it already covered specifically in the initial AcrIIA paper? Is anything gained here that wasn't in figure 1A?

5.8 Fig 8. I feel this figure needs to be re-evaluated – what is it actually presenting? First of all, there is no legend for what colour coding means (what % similarity. AA? Nucleotide? Scale for all of them? Function? Domain only?). Secondly, at a glance at the figure I don't gain much information other than there are uncharacterized Acrs across diverse species, generally near HTH-motif Acas. It's not

really helpful that some are referred to as HTH only, others Aca3 homologs. Panel B is misleading; RcapNL isn't a separate entity from the strains, its prophage in those strains – shouldn't these just be one MGE – RcapNL (present in DE442, Y262, R121?). I wished the Figure itself had information about the likely CRISPR-Cas systems targeted rather than finding this in the candidates sections only.

Minor:

Some more information on the 7 known ACR families not in your database and why (what criterion lost them? A 17.9% loss is high, although justifiable if you give the justification

p19 Not clear how the Top 30 families are defined, which makes p20 – 22 challenging to read as well.

Line-by-Line (Without line numbers, but in order):

-Title: Vast Diversity is a little wishy-washy. Avoid adverb? Title could probably benefit from highlighting the new families (not just new proteins)

- Abstract (and elsewhere). Numerous, diverse / numerous, diverse / small, highly variable – many of these adjective, [adv] adjective constructions in a short space – and throughout the manuscript that create poor stylistic repetition.

- Abstract: prediction. Bioinformatic?

- Abstract: An examination... Acr features. A few gripes here – one is that the model was based on these features, so this is a bit tautological. It might help to list these features in the abstract.

- Abstract: One of the top candidates... independently tested. I understand what is meant, but this read like it was part of this paper, *and* calling it independent when there is a shared author between the two works is a bit of a stretch. Consider rewording – not entirely sure how. Hard to have this in the abstract without it seeming like its part of the paper. “Coincidentally, an other approach has since validated ?”

-Intro: nearly all/ - refer to Burstein 2016 paper (PMID: 26837824), or possibly Bernheim 2020 (31745554). I am always wary of leaning too heavily on these, comparatively old, proportions of CRISPR in bacteria. Is there a need to cite both 2 and 4?

- Intro: variability ; consider (briefly) highlighting how diverse the systems they interfere with are, which is at least part of the reason they themselves are so diverse.
- Intro: The two primary ones = the two primary bioinformatic ones, as in-vivo screens have been successfully at finding ACRs.
- Intro: Acrs themselves which (is missing a comma)
- Intro: encode functional CRISPR-Cas (predicted to be functional?)
- Intro: The main challenge.... Formidable challenge – this section feels very repetitive both internally , and to the ending of the first paragraph.
- Intro: significantly predictive – is this a term of art? I am always wary of the word significant without an immediately referenced test.
- Intro: examine in detail / examine the top candidates in detail.

Results:

- “strive” – very odd to have this in present tense (a couple other cases too, when most of the rest of the results are in past tense)
- Throughought: us of “(see methods for details)” – this is always assumed. Please omit.
- “Psi-BLASTed” = informal lab-speak (e.g. PCRd vs performed a PCR).
- “ACRs are also typically encoded”—this is in the middle of your presentation of results, so it feels like you are presenting your findings, but the reference you cite established this long ago. F
- “Gave any significant hits”: reword/specify test, etc
- “domains from either CDD or pVOG” : Unclear here if a hit to CDD or pVOG is weighted against or for the Acrs (this is clarified later, please state here)
- “Has been published” : why this tense?
- “It has less variance and is therefore less likely to overfit”: this reads a little informal?
- “Gini impurity”: I am not familiar with this term.
- “(this can be any value between... , inclusive)” this can be omitted, the reader can be expected to understand fractions.
- “large class imbalance”? What is this?
- “ROC/AUC analysis”. For a more lay reader, consider defining the metrics (True positive and false positive rates) - then, if they don’t know what an ROC is, they will still understand the purpose of the model.
- “a genetic algorithm” ? what is this?

- “Significantly predictive of Acrs with an AUC” - should have a comma between Acrs and with.
- “predictive of novel, heretofore unseen Acrs” (not clear how different those Acrs in the new set really are, or how novel they are – this is not established. Consider constraining language a bit to predictive, for a set, of heretofore unseen Acrs)
- “Among which 20 included” = awkward. This structure hurts also a little further down – its not clear which 20, or which 2 are lost with the directon filter
- (predicted) proviruses: predicted how/ by what tools.
- “Similarity to known Acrs are homologous to ... (OrfA, B, E)” – omit the parenthetical, it does nothing to help and just confused me. You can touch on the Orfs later when the figure presents them.
- “did not differ significantly from the negative set” (statistical test?)
- “Acr/CRISPR distribution”. Why only list top 3? I bet people want to know about IIA
- Not clear if the “Among the Candidate ACR” paragraphs counts of single/2/3 or more are only of the novel acrs , or total acrs, .
- “predicted acrs predicted for” = awkward sentence.
- “Within these neighborhoods, we also find” (omit comma)
- p19 Paragraph beginning with Here, we present is re-treading a lot of ground already covered, and
- First paragraph of the discussion was again, a very very close recap of material already presented.
- References inconsistent as to whether in title case or sentence case.

Signed: Alexander P. Hynes (I was delighted to see my painstakingly annotated RcapNL prove useful!)

Reviewer #2 (Remarks to the Author):

Summary:

Anti-CRISPR proteins (Acrs) can inhibit CRISPR-Cas and thus have potential for application as modulators of genome editing tools. Since most Acrs are small and highly variable proteins,

prediction of Acrs is a challenge. The authors develop a machine learning approach for 'comprehensive' Acr prediction and present ~2,500 novel candidate Acr families. Additional examination of the top candidates confirmed they possess typical Acr features, and one of the top candidates (i.e. AcrIIA12) was independently tested and found to possess anti-CRISPR activity by another group (Osuna, B.A. et al, 2019, BioRxiv). The authors provide a web resource to access their predicted Acr sequences, allowing researchers to explore the function of these Acrs in a variety of different organisms.

Comments:

The authors present a nice approach that provides a useful resource for future studies looking to experimentally validate and employ Acrs in gene editing applications. The one issue with the manuscript is that the Acr families that are predicted are highly biased towards those that resemble known Acr families. The model has blind-spots, which is inherent to the study design and there is not much they can do to address this other than provide a more comprehensive description of their model performance. The manuscript is clear and thorough, and the authors clearly went to lots of trouble to choose a good negative training set (i.e. non-Acrs) for developing their predictive model. A few points come to mind worth considering.

1. Although the authors state that the choice of using random forests is avoid over-fitting, it is not clear why they could not try other methods like logistical regression or convolutional neural nets for benchmarking purposes? Running these approaches is not difficult once the training sets have been built.
2. Error-analysis: the final model has an AUC=0.83 when evaluated using the test set. For classifiers like this it helps to understand what the characteristic of the incorrectly classified Acrs are. Are the incorrect classifications randomly distributed across Acr classes? Or is there a certain family of Acrs that are systematically misclassified?
3. Reporting on performance metrics: AUC does not provide a complete picture of the model's performance and additional metrics should be reported, especially since the focal point of the study is the model that they developed. Specifically, true positive/true negative predictions, false negatives, and F1 metrics are usually discussed. These metrics will provide additional clarification on how well the model performs.

Another important realization is that the AUC = 0.83 that is reported is not really what they achieve when predicting novel Acrs because the authors apply a handful of additional heuristic filters post-prediction which can skew the performance of the model beyond what was initially characterized. These heuristic filters were based off of what is known from previously characterized Acrs, introducing additional biases and potentially overlooking Acr families that are categorically distinct

from those that have been characterized to-date. The authors did allude to this as a potential caveat in the discussion (“the possibility exists that, using the approach described here, we only detect one, albeit major, class of Acrs, whereas others might exhibit distinct properties”), which may warrant further explanation.

We thank the reviewers for the constructive, helpful comments which are all addressed in the revised manuscript as detailed below. In addition to the revisions made in response to the reviewers' comments, we now have limited but, we believe, important results on experimental validation of our predictions that have been included in the revised manuscript as a separate section under Results (lines 449-478), the new Figure 7 and Supplementary Table 6. This addition to the work necessitated inclusion of two new authors who performed the experiments. In our view, this substantially strengthened the work.

Reviewer #1 (Remarks to the Author):

In this work, Gussow et al present a new, machine-learning-based approach to bioinformatic prediction of anti-CRISPRs. The result is an enormous expansion of Acrs and Acr families. Furthermore, as these are fascinating (and commercially exploitable) proteins, the work is impactful and an important contribution to a hot and fast-moving field. It is thorough, too – catching tricky concepts like the error in sequence annotation for AcrIIA4s. Great work!

We appreciate these positive, encouraging comments.

I have no critical or major concerns. The paper is well suited for the journal, presents a major advance in the field, and does not overextend its findings. However, I did find the paper challenging to read. While the flow of ideas was logical and the language used was appropriate, there was a lot of repetition of specific phrases that felt like padding, which detracted from the manuscript. I also found the Figures to be challenging to navigate, often for reasons with very simple fixes – and almost all could have used more detailed legends (and moved below the figures for legibility). I will begin by listing a few Moderate and Minor concerns and append a list of line-specific issues.

I do apologize for the emphasis on flow/wording. In this lockdown mode with 3 young children at home, my review time is limited and issues like these disproportionately stand out.

We greatly appreciate the reviewer diligent attention to our manuscript under these unusual conditions. The suggestions on the improvement of the presentation are valuable.

Moderate:

(1) This point concerns me throughout the manuscript, and it probably has more to do with my only passing familiarity with bioinformatics. The dataset was built eliminating hits to large proteins, but then the supposed striking feature of ACRs is their small size - how influenced is this by the bias inserted at that earlier stage?

In constructing the positive training set, we sought to find representatives in our data of true Acrs. To this end, we ran PSIBLAST against our dataset. One common type of false positive is when a smaller protein, such as an Acr, by chance has some similarity to a portion of a very large protein leading to a false positive hit. We therefore eliminated such hits, to avoid biasing the training set with non-Acrs.

This was not clear in the original text, and has been edited accordingly, lines 86-91.

Many ACRs were identified next to Acas, so how surprise is it really that your Acr database also includes ACRs typically encoded upstream of Acas?

This is not surprising and indeed expected given the training set. We intended the statement as an affirmation that this known characteristic occurs in our set. However, to avoid any ambiguity, we have removed this line from the text.

Later, you identified a subset of the prokaryotic database with complete CRISPR-Cas systems – is this not stacking your dataset for self-targeting spacers? Shouldn't the algorithm handle these and sort away the non-CRISPRs anyway (since self-targeting is weighted), rather than biasing the search dataset?

We appreciate the comment, this was indeed unclear in the original manuscript. There are two stages in this work, of which the first stage involves creating a model that can predict Acrs, and assigns higher scores to Acrs than non-Acra. This is the stage we assessed using the test set. The second stage is creating a dataset of proteins enriched for Acra. This stage is not meant to assess the model formally (as this was already done in the previous section), but rather to generate a set enriched for Acra by combining the model with other enrichment approaches. We have clarified this in the revised text, lines 203-208.

Then, you filter the set to remove those with significant (? Unclear on criterion) hits to CDD or pVOG. Again, this is one of your machine learning criteria anyway, so why do you need to bias the dataset? As stated, I feel this is likely just a failure on my end to grasp some simple element, but as this feeling pervaded my entire readthrough of the manuscript, it may be important to address.

Apparently, there was lack of clarity on this in the original manuscript; per our response to the previous comment, the idea here was not to formally assess the model, as we did in the earlier stage of the analysis, but rather, to generate an enriched set by combining the model with other enrichment procedures, including the removal of the hits to domain databases and the heuristics. This has been clarified in the revised text accordingly, lines 203-208. The criterion was clarified in the methods section, lines 554-556.

(2) You removed families with numerous homologs in diverse prokaryotes. I worry this biases against acra in bacteria over-represented in the sequence databases? I doubt there is a real issue here, as it's not like your cutoff of 375 was anything close to your largest family (319).

We found that overall Acra in our positive set tend to have few homologs, and therefore, this cutoff was set to eliminate likely false-positives that are widespread. However, we acknowledge that the heuristic approach can lead to biases in the resulting set and have reiterated this in the revised text, lines 205-207 and lines 265-267.

(3) I... think this may be a stupid question, but how many of your training set would survive your heuristics from Fig 3? Might this be worth having as a panel for Fig 3 as well? I am almost certain this is a stupid question.

This is a good question. Initially, 33/232,616 of the clusters in our set included Acra from our positive set, compared to 17/2,526 of the final set. We have added this to the revised text and to Figure 3, and included a new supplemental file with this information (Figure 3b, Supplementary Table 3 and lines 262-263).

(4) Is the resolving power between a 3.49 and 3.5 mean directon size really useful? That seems... wild.

The means are meant to provide a general idea of the distributions of Acrs / non-Acra, however, the model does not use these as a cutoff. Rather, the random forest can model complex interactions between features, so if features have close or even the same means or even similar distributions, the direction size interaction with other features could provide useful information. We have added this to the random forest explanation, lines 151-154 and lines 617-619.

(5) The figures need improvements throughout the manuscript. In some cases they should be omitted (Fig 7), most – more carefully labelled. Here are some specific points on the figures

5.1 Fig 1 axes are confusing – protein length is in what units? Are Direction Protein lengths Mean the average length of proteins in the direction? Presumably, but this could be clearer. Is density the correct Y label? I assume it is count?

The labels have been updated and an explanation has been added to the legend. The Y axis is the probability density function, a smoothed continuous function estimated from the counts data. This has been added to the figure legend, lines 117-122.

5.2 Fig 2: why the red dot? Why is this not a line, a threshold... why is it at 0.5 on the y axis?

The red dot was meant to indicate the true AUC in order to compare to the empirical null distribution. To avoid this confusion, the red dot was converted to a red line, and the figure legend updated, lines 191-192.

5.3 I think the paper might benefit from emphasizing the new criteria and heuristics – in the tables 1 and 2, for instance. IN table 2 the first 2 criteria are the ones everyone uses while the next 4 are new.

We now explicitly differentiate the thresholds that were previously established and those introduced here, in the main text, lines 239-240 and in the Figure 3 legend, lines 275-277.

I also think Figure 3 and Table 2 should be merged together

Figure 3 and Table 2 have been merged accordingly.

5.4 Fig 4: Legend should at least give us cursory info about families with more than 25 members (the X families with more than 25 members). Better axis labels; Size of family (# of members)

The axis label has been updated accordingly and we have included cursory information about families with more than 25 members, lines 297-298.

5.5 Fig 5: show mean on figures. Show units (aa)

The mean and the units have been added to Figure 5.

5.6 Fig 6 is hard to read. Grey in legend is white? I bet you can do more with this figure.

Thank you for this comment. Figure 6 has been edited to include axes and have better contrast.

5.7 Fig 7 does not feel like it should be its own figure. Isn't it already covered specifically in the initial AcrIIA paper? Is anything gained here that wasn't in figure 1A?

We agree with the reviewer and Figure 7 has been removed.

5.8 Fig 8. I feel this figure needs to be re-evaluated – what is it actually presenting? First of all, there is no legend for what colour coding means (what % similarity. AA? Nucleotide? Scale for all of them? Function? Domain only?). Secondly, at a glance at the figure I don't gain much information other than there are uncharacterized Acra across diverse species, generally near HTH-motif Acra. It's not really

helpful that some are referred to as HTH only, others Aca3 homologs. Panel B is misleading; RcapNL isn't a separate entity from the strains, its prophage in those strains – shouldn't these just be one MGE – RcapNL (present in DE442, Y262, R121?). I wished the Figure itself had information about the likely CRISPR-Cas systems targeted rather than finding this in the candidates sections only.

We agree with the reviewer and Figure 8 has been removed accordingly.

Minor:

Some more information on the 7 known ACR families not in your database and why (what criterion lost them? A 17.9% loss is high, although justifiable if you give the justification

We believe that this is indeed a justifiable loss that warrants additional comment. As the database used in this study was curated in 2016 and does not include all known proteins, this loss is not surprising. Additionally, Acrs are fast evolving and tend to be highly variable with few homologs, therefore are likely to be underrepresented in sequence databases. All of the 7 families not in our database were originally detected in strains that were not available at the time the database was constructed. Reconstruction of the database is not within the scope of this paper. This justification has been added to the revised text, lines 531-535.

p19 Not clear how the Top 30 families are defined, which makes p20 – 22 challenging to read as well. This set was created by filtering for clusters with more than 4 members selecting the 30 clusters with the highest mean model score. This has been clarified in the text, lines 370-372.

Line-by-Line (Without line numbers, but in order):

-Title: Vast Diversity is a little wishy-washy. Avoid adverb? Title could probably benefit from highlighting the new families (not just new proteins)

The title has been edited accordingly to:

Novel anti-CRISPR protein families predicted with a machine-learning approach

- Abstract (and elsewhere). Numerous, diverse / numerous, diverse / small, highly variable – many of these adjective, [adv] adjective constructions in a short space – and throughout the manuscript that create poor stylistic repetition.

These instances in the abstract and elsewhere have been edited accordingly, lines 15-29.

- Abstract: prediction. Bioinformatic?

We have clarified that the prediction described was bioinformatic, lines 20-21.

- Abstract: An examination... Acr features. A few gripes here – one is that the model was based on these features, so this is a bit tautological. It might help to list these features in the abstract.

We agree with the reviewer, and for clarity, we have removed this sentence, lines 15-29.

- Abstract: One of the top candidates... independently tested. I understand what is meant, but this read like it was part of this paper, *and* calling it independent when there is a shared author between the two works is a bit of a stretch. Consider rewording – not entirely sure how. Hard to have this in the abstract without it seeming like its part of the paper. "Coincidentally, an other approach has since validated?"

The abstract has been reworded as suggested, lines 25-26.

-Intro: nearly all/ - refer to Burstein 2016 paper (PMID: 26837824), or possibly Bernheim 2020 (31745554). I am always wary of leaning too heavily on these, comparatively old, proportions of CRISPR in bacteria. Is there a need to cite both 2 and 4?

We added the Burstein 2016 paper and deleted the 2011 reference, lines 34-36.

- Intro: variability ; consider (briefly) highlighting how diverse the systems they interfere with are, which is at least part of the reason they themselves are so diverse.

We now mention the diversity of these systems, lines 47-49.

- Intro: The two primary ones = the two primary bioinformatic ones, as in-vivo screens have been successfully at finding ACRs.

We have edited the text accordingly, lines 51-53.

- Intro: Acrs themselves which (is missing a comma)

We have edited the text accordingly, lines 56-58.

- Intro: encode functional CRISPR-Cas (predicted to be functional?)

We have edited the text accordingly, lines 61-62.

- Intro: The main challenge.... Formidable challenge – this section feels very repetitive both internally , and to the ending of the first paragraph.

We have edited the text to avoid these repetitions, lines 47-53.

- Intro: significantly predictive – is this a term of art? I am always wary of the word significant without an immediately referenced test.

This is indeed about statistical significance which is explicitly indicated under Results (lines 183-184, and lines 196-197). However, this seemed to be out of place in the Introduction. To the best of our understanding, using 'significant' is permissible as long as statistical significance is actually meant (which is our case) but immediate indication of the p-value and the test is not strictly mandatory.

- Intro: examine in detail / examine the top candidates in detail.

We have edited the text as suggested, lines 72-74.

Results:

- “strive” – very odd to have this in present tense (a couple other cases too, when most of the rest of the results are in past tense)

We have edited the text as suggested, lines 78-79.

- Throughought: us of “(see methods for details)” – this is always assumed. Please omit.

We have removed most instances of references to Methods in this section.

- “Psi-BLASTed” = informal lab-speak (e.g. PCRd vs performed a PCR).

We have corrected this use throughout the manuscript, lines 86-88.

- “ACRs are also typically encoded”—this is in the middle of your presentation of results, so it feels like you are presenting your findings, but the reference you cite established this long ago. F

We have removed this sentence.

- “Gave any significant hits”: reword/specify test, etc

We have reworded this sentence accordingly, lines 102-106.

- “domains from either CDD or pVOG” : Unclear here if a hit to CDD or pVOG is weighted against or for the Acrs (this is clarified later, please state here)

We have added the expectation that proteins with conserved domains likely perform other functions and therefore are less likely to be Acrs, lines 102-106.

- “Has been published” : why this tense?

We have edited accordingly, line 142.

- “It has less variance and is therefore less likely to overfit”: this reads a little informal?

We have expanded and clarified this sentence, lines 151-154.

-“Gini impurity”: I am not familiar with this term.

We have reworded this sentence for clarity, line 159, and described the Gini impurity calculation in the methods, lines 622-629.

-(“this can be any value between... , inclusive)” this can be omitted, the reader can be expected to understand fractions.

We have accordingly removed this from the text.

-“large class imbalance”? What is this?

This has been reworded and expanded for clarity, lines 171-174. Class imbalance occurs when one set is larger than another. In this case, the positive set is smaller than our negative set.

-“ROC/AUC analysis”. For a more lay reader, consider defining the metrics (True positive and false positive rates) - then, if they don’t know what an ROC is, they will still understand the purpose of the model.

The explanations for TPR / FPR have been added to the text, lines 177-179.

-“a genetic algorithm” ? what is this?

A genetic algorithm is a feature selection algorithm that selects subsets of features and creates different feature combinations while optimizing for the best feature set. These combinations are created over several iterations, which are referred to as generations. A reference has been added along with an explanation in the Methods section, lines 602-604.

-“Significantly predictive of Acrs with an AUC” - should have a comma between Acrs and with.

We have edited the text accordingly, line 184.

-“predictive of novel, heretofore unseen Acrs” (not clear how different those Acrs in the new set really are, or how novel they are – this is not established. Consider constraining language a bit to predictive, for a set, of heretofore unseen Acrs)

We have edited the text to clarify that the model is predictive of Acrs that were not present in the training set, lines 184-185.

-“Among which 20 included” = awkward. This structure hurts also a little further down – its not clear which 20, or which 2 are lost with the directon filter

We have edited the text accordingly, lines 236-237, and this information is now included in Figure 3.

-(predicted) proviruses: predicted how/ by what tools.

This has been rewored to indicate that they were not predicted proviruses, rather putative proviruses detected as described in the next few sentences, line 217.

-“Similarity to known Acrs are homologous to ... (OrfA, B, E)” – omit the parenthetical, it does nothing to help and just confused me. You can touch on the Orfs later when the figure presents them.

We have removed the parenthetical and added a clarification for these Orfs, lines 289-290.

-“did not differ significantly from the negative set” (statistical test?)

We appreciate this comment, ‘significantly’ was inappropriate here without a statistical test. After running a Mann-Whitney U test, we found that the percent of beta sheets differs with statistical significance, though not substantially, with 13% in the Acr candidates and 15% in the negative set. We corrected the language here to reflect this and added the details of the statistical test, along with two box plots (Supplementary Figure 2) illustrating these differences, lines 316-323.

-“Acr/CRISPR distribution”. Why only list top 3? I bet people want to know about IIA

We’ve clarified this section, and added a Supplementary File with the proportion of genomes that include each subtype, including II-A (Supplementary Table 4), lines 334-337.

-Not clear if the “Among the Candidate ACR” paragraphs counts of single/2/3 or more are only of the novel acrs , or total acrs, .

The text has been edited to clarify that this applies to the 2,500 predicted Acrs, line 339.

-“predicted acrs predicted for” = awkward sentence.

We have edited the text accordingly, lines 342-343.

-“Within these neighborhoods, we also find” (omit comma)

We have edited the text accordingly, line 357.

- p19 Paragraph beginning with Here, we present is re-treading a lot of ground already covered, and

We removed this paragraph accordingly.

-First paragraph of the discussion was again, a very very close recap of material already presented.

This paragraph was shortened, lines 481-483.

-References inconsistent as to whether in title case or sentence case.

Thank you for noticing this, the references were converted to sentence case.

Signed: Alexander P. Hynes (I was delighted to see my painstakingly annotated RcapNL prove useful!)

Reviewer #2 (Remarks to the Author):

Summary:

Anti-CRISPR proteins (Acrs) can inhibit CRISPR-Cas and thus have potential for application as modulators of genome editing tools. Since most Acrs are small and highly variable proteins, prediction of Acrs is a challenge. The authors develop a machine learning approach for 'comprehensive' Acr prediction and present ~2,500 novel candidate Acr families. Additional examination of the top candidates confirmed they possess typical Acr features, and one of the top candidates (i.e. AcrIIA12) was independently tested and found to possess anti-CRISPR activity by another group (Osuna, B.A. et al, 2019, BioRxiv). The authors provide a web resource to access their predicted Acr sequences, allowing researchers to explore the function of these Acrs in a variety of different organisms.

Comments:

The authors present a nice approach that provides a useful resource for future studies looking to experimentally validate and employ Acrs in gene editing applications. The one issue with the manuscript is that the Acr families that are predicted are highly biased towards those that resemble known Acr families. The model has blind-spots, which is inherent to the study design and there is not much they can do to address this other than provide a more comprehensive description of their model performance. The manuscript is clear and thorough, and the authors clearly went to lots of trouble to choose a good negative training set (i.e. non-Acrs) for developing their predictive model.

We appreciate these encouraging comments.

A few points come to mind worth considering.

1. Although the authors state that the choice of using random forests is avoid over-fitting, it is not clear why they could not try other methods like logistical regression or convolutional neural nets for benchmarking purposes? Running these approaches is not difficult once the training sets have been built.

The selection of random forest was performed carefully given the nature of the data, this was, indeed, not made sufficiently clear in the original text.

Given the small amount of data available to us, a convolutional neural net would not have been appropriate, as these require much larger amounts of data given their complexity and large number of parameters to estimate. A logistic regression model would be more appropriate if we had known from the start which features are important, and if we knew that the data is linearly separable by those features, so that they can linearly divide the Acrs from the non-Acrs. However, by using a random forest, we can allow non-linear classifier generation using this data, and model, to some extent, interactions between features, with low risk of overfitting given that it is an ensemble classifier. For these reasons, we chose to use the random forest model.

The motivation behind the selection of a random forest model has now been clarified and added to the text, lines 151-154 and lines 617-619.

2. Error-analysis: the final model has an AUC=0.83 when evaluated using the test set. For classifiers like this it helps to understand what the characteristic of the incorrectly classified Acrs are. Are the incorrect classifications randomly distributed across Acr classes? Or is there a certain family of Acrs that are systematically misclassified?

We appreciate this comment. Members of three of the six Acr families assessed in the test set were detected most of the time (AcrIF12-IF14), whereas the members of the remaining three families were detected less than half of the time, with the single member of AcrIE7 in the test set not detected by the model. This has been added to the text, lines 197-200, and we have added Supplementary Figure 1d and Supplementary Table 2 containing this data.

3. Reporting on performance metrics: AUC does not provide a complete picture of the model's performance and additional metrics should be reported, especially since the focal point of the study is the model that they developed. Specifically, true positive/true negative predictions, false negatives, and F1 metrics are usually discussed. These metrics will provide additional clarification on how well the model performs.

We have added a section assessing the model's binary predictions, including the precision (fraction of proteins predicted as Acrs that are Acrs) and recall (fraction of the Acrs that are correctly classified), as the F1 score is the mean of the precision and recall. We also added Supplementary Figure 1, including the threshold selection and a comparison of the precision and recall to random permutations of the data labels, lines 194-200.

Another important realization is that the AUC = 0.83 that is reported is not really what they achieve when predicting novel Acrs because the authors apply a handful of additional heuristic filters post-prediction which can skew the performance of the model beyond what was initially characterized. These heuristic filters were based off of what is known from previously characterized Acrs, introducing additional biases and potentially overlooking Acr families that are categorically distinct from those that have been characterized to-date. The authors did allude to this as a potential caveat in the discussion ("the possibility exists that, using the approach described here, we only detect one, albeit major, class of Acrs, whereas others might exhibit distinct properties"), which may warrant further explanation.

We have added this caveat to the first paragraph of the section for novel Acr prediction, lines 203-208.